# SLIM-Brain: A Data- and Training-Efficient Foundation Model for fMRI Data Analysis

## Abstract

Foundation models are emerging as a powerful paradigm for fMRI analysis, but current approaches face a dual bottleneck of data- and training-efficiency. Atlas-based methods aggregate voxel signals into fixed regions of interest, reducing data dimensionality but discarding fine-grained spatial details, and requiring extremely large cohorts to train effectively as general-purpose foundation models. Atlas-free methods, on the other hand, operate directly on voxel-level information - preserving spatial fidelity but are prohibitively memory- and compute-intensive, making large-scale pre-training infeasible. We introduce **SLIM-Brain** (**S**ample-efficient, **L**ow-memory f**MRI** Foundation **M**odel for Human **Brain**), a new atlas-free foundation model that simultaneously improves both data- and training-efficiency. SLIM-Brain adopts a two-stage adaptive design: (i) a lightweight temporal extractor captures global context across full sequences and ranks data windows by saliency, and (ii) a 4D hierarchical encoder (Hiera-JEPA) learns fine-grained voxel-level representations only from the top-$k$ selected windows, while deleting about 70% masked patches. Extensive experiments across seven public benchmarks show that SLIM-Brain establishes new state-of-the-art performance on diverse tasks, while requiring only 4 thousand pre-training sessions and approximately 30% of GPU memory comparing to traditional voxel-level methods. Code and trained weights of SLIM-Brain are available at `https://anonymous.4open.science/r/SLIM-Brain-9C51`.

## 1 Introduction

Functional Magnetic Resonance Imaging (fMRI) has been the de facto modality for non-invasive analysis of human brain activities, with broad applications from clinical diagnostics, to monitoring neurological conditions and understanding cognitive processes (Song et al., 2008; Horikawa & Kamitani, 2017). Modern MRI scanners capture brain activity via monitoring the Blood Oxygen Level Dependent (BOLD) signals at voxel level of human brains, and are capable of acquiring high-resolution volumetric data (e.g. up-to $1mm$ spatial resolution) over time. As a result, a single scan can generate a massive four-dimensional (4D) data sequence (3D space×time), posing significant challenges to extract meaningful representations of brain activities, functional connectivity, and their associations with behavior and diseases.

Instead of directly processing the massive volumetric fMRI data, existing studies often rely on atlas-based parcellations, where voxel-level signals are aggregated within template-defined anatomical brain regions - referred to as Regions of Interest (ROIs) in the following - according to an "atlas", effectively

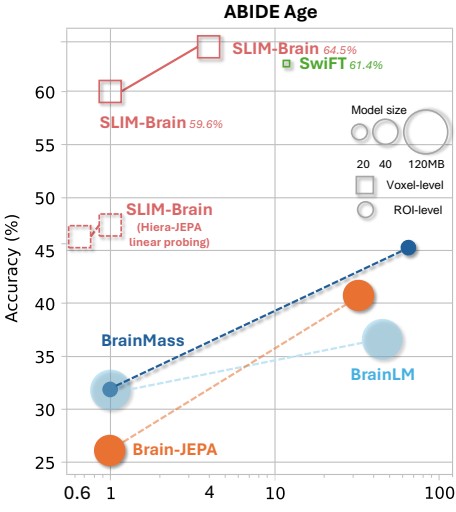

Figure 1: **Performance & Pretraining size.** Our method *SLIM-Brain* reaches 64.5% age-classification accuracy with only about **4 thousand** sessions in pretraining.

Figure 2: **(a) ROI-based.** Atlas parcellation coarsely downsamples space, introducing atlas bias and erasing voxel-level detail. **(b) Volume sliding-window pipelines.** Resolution is retained, but fixed window lengths (e.g., 40 frames) with simple averaging dilute transient events and miss cross-window dynamics. **(c) Ours.** A lightweight global pass ranks windows; the top small windows (e.g., 5 frames) are concatenated to a set (e.g., 40 frames) and encoded with a 4D encoder and fused with global features, yielding efficient multi-granularity representations with fine spatial semantics and long-range spatiotemporal structure.

converting the 4D data into a lower dimensional 2D format: i.e., ROIs ×time (Fig. 2a). They then apply signal processing and/or machine learning models tailored for specific tasks on such 2D data, e.g. disease classification (Sidhu et al., 2013; Arbabshirani et al., 2017). However, these supervised approaches typically require labeled data to train their models, while recent studies (Marek et al., 2022) have shown that to achieve statistically reliable results, very large cohorts (e.g. often >1000 participants) are necessary - adding yet another layer of challenges in practical fMRI analysis.

More recently, there has been a growing interest in developing deep learning based *foundation models* for fMRI data analysis, inspired by their remarkable performance in Computer Vision and Natural Language Processing tasks. The idea is to pre-train (usually in a self-supervised way) a large model to learn general-purpose representations of brain activity on vast neuroimaging data, which can then be adapted to diverse downstream tasks with limited labeled data (e.g., via fine-tuning). Broadly speaking, current efforts on building foundation models, or more generally applying deep learning techniques in fMRI data analysis can be categorized into two threads: *atlas-based* and *atlas-free* approaches. The former continues the traditional paradigm of summarizing voxel-level signals into predefined brain regions (Caro et al., 2023; Assran et al., 2023; Yang et al., 2024), therefore leveraging anatomical priors and improving interpretability. However those methods bear several key limitations: i) there is no universally optimal atlas: performance on different downstream tasks may highly depend on parcellation choices, where results across studies using heterogeneous atlas are not directly comparable (wang et al., 2025; Salehi et al., 2020); ii) averaging based on any pre-defined atlas will inevitably discard important voxel-level information - as a result such models often need to be trained on very large cohorts (e.g., $\sim 60k$ sessions) to perform well (Dong et al., 2024; Caro et al., 2023) (see Fig. 1); and iii) any analysis using such models will be confined to the parcel resolution of the atlas used, where probing within-region structures (e.g., isolating amygdalar subnuclei during fear conditioning) is impossible (Wen et al., 2022).

On the other hand, atlas-free methods (Zhao et al., 2018; Vu et al., 2020; Nguyen et al., 2020) aim to learn directly from voxel-level data without imposing ROI boundaries of certain atlas, allowing the learned models to capture fine-grained functional patterns and potentially discover novel brain organization. However, existing atlas-free approaches have typically been developed for specific tasks rather than as general-purpose foundation models, due to their prohibitively high training cost. The deep learning architectures underlying those models, such as the widely adopted Vision Transformers (ViT), incur quadratic cost (both memory and compute) with respect to input dimensions. This makes it nearly impossible to pre-train them on large-scale fMRI data, and thus building atlas-free foundation models still remains an open challenge. Recent work has explored efficient variants such as Shifted-window (Swin) Transformer (Kim et al., 2023; Sun et al., 2025; Kwon et al., 2024; Peng et al., 2025b), but at each timestamp they still feed the entire dense fMRI volume into the encoder, wasting resources on those voxels ($\approx 70\%$) outside the brain with no valid signal. Similar inefficiencies also arise along the temporal axis, where existing approaches either only train with data pertain to specific task states, e.g., extracting $\sim 30$ timestamps (Shi et al., 2023; Sun et al., 2025; Rosenman et al., 2024), or consider sliding windows with limited sizes, where data within each window is processed independently through the model, with results aggregated thereafter (Kim et al., 2023)(e.g.,

as shown in Fig. 2b). In practice neither is ideal: the former obviously hurts generalization capabilities of the model beyond task states, while the latter evenly processes every window - may lose focus on those truly important data segments - leading to inferior performance comparing to some of the recent atlas-based approaches in out-of-distribution cases (Dong et al., 2024).

To address these challenges, in this paper we propose **SLIM-Brain**, a new atlas-free foundation model for fMRI analysis that overcomes the limitations of both existing atlas-based and atlas-free approaches. SLIM-Brain achieves the best of both worlds by jointly pushing data and training efficiency: it requires much less data during pre-training - outperforming state-of-the-art atlas-based models with only a fraction of their training data, while at the same time reducing memory/compute by an order of magnitude compared to the most recent atlas-free methods. At its core, SLIM-Brain adopts a novel two-stage adaptive paradigm, working in tandem across the temporal and spatial domains: i) a lightweight temporal extractor that performs coarse sweeps over full sequences, capturing global context and identifying the top-$k$ most informative data windows as shown in Fig. 2c; and ii) an efficient encoder based on hierarchical Joint Embedding Predictive Architecture (Hiera-JEPA) that delves into the selected windows, but only focusing on voxels with valid signals rather than processing the full volumes. Concretely, the technical contributions of this paper are as follows:

- To the best of our knowledge, we are the first to systematically study the dual challenges of data efficiency (the heavy reliance on extremely large cohorts) and training efficiency (prohibitively high memory and compute costs) in foundation models for fMRI analysis, arguing that these are the primary obstacles preventing the widespread adoption of such models in practical settings.

- We propose SLIM-Brain, a data- and training-efficient foundation model built upon a two-stage adaptive pipeline. First, a lightweight extractor captures coarse global context and ranks small temporal windows. Next, only the selected data windows are concatenated and processed by a hierarchical Joint Embedding Predictive Architecture (Hiera-JEPA) encoder, which focuses exclusively on voxels with valid signals, discarding approximately 70% masked patches. This adaptive design yields fine-grained voxel-level features while reducing GPU memory usage to about 30% of Swin-based models.

- We validate SLIM-Brain extensively across multiple downstream tasks (e.g. sex, age and fingerprint classification) and four different datasets. Results show that SLIM-Brain establishes the new state-of-the-art performance on diverse tasks, while requiring only a tiny amount of data ($1k$ vs. $32k$ sessions) in pre-training than the strongest baselines.

## 2 RELATED WORK

**Foundation Models for fMRI Analysis.** fMRI signals reflect ongoing brain states and cognitive processes. Early work framed decoding as supervised classification or regression on activity patterns, which often produced task-specific features with limited out-of-distribution generalization (Song et al., 2008; Horikawa & Kamitani, 2017; Ye et al., 2023; Zhao et al., 2018; Vu et al., 2020; Nguyen et al., 2020). More recently, the field has pivoted to task-agnostic foundation models trained with self-supervised pipelines on unlabeled data, aiming for representations that transfer across tasks and datasets. One prominent line of work pools voxel signals within a predefined atlas to obtain region-wise time series and then learns objectives on those ROI tokens—either generative (masked reconstruction, as in BrainLM) or latent prediction (as in Brain-JEPA) (Caro et al., 2023; Dong et al., 2024). These approaches are memory-efficient and scalable, but their quality can depend on the chosen parcellation and its granularity. A related direction converts the atlas-reduced time series into pairwise or higher-order functional graphs and optimizes unsupervised objectives on these structures (Yang et al., 2024; Thapaliya et al., 2024; Han et al., 2025). For example, BrainMass augments networks by randomly dropping time points from the BOLD signal during training to encourage robustness (Yang et al., 2024), and hypergraph formulations have been explored to capture higher-order relationships (Han et al., 2025). Complementing these atlas-based routes, volumetric encoders operate directly on 4-D fMRI volumes, thereby avoiding atlas-induced bias and preserving fine spatial detail (Peng et al., 2025a; Sun et al., 2025; Kwon et al., 2024; Kim et al., 2023).In practice, computational and memory constraints often necessitate training on short, task-aligned excerpts rather than full-length recordings. Typical applications include task classification (Shi et al., 2023),

brain-state decoding (Sun et al., 2025), and stress prediction (Rosenman et al., 2024). Turning these systems into a general-purpose foundation model remains a major challenge.

**Efficient Vision Transformers.** Vanilla Vision Transformers (ViTs) apply global self-attention over all tokens. The cost scales quadratically with token count and the model offers weak spatial inductive bias (Dosovitskiy et al., 2020). This is impractical for high-resolution or long 4-D fMRI sequences, where a single scan can yield millions of voxel tokens across time. Hierarchical Transformers address these issues by progressively downsampling tokens and widening channels (e.g., MViT, Swin) (Fan et al., 2021; Liu et al., 2021). For example, Swin restricts attention to local windows and cyclically shifts the partition between layers, which stabilizes memory and accuracy on dense inputs. However, it limits global context to multi-layer message passing, ties the model to a dense, regular lattice, and introduces engineering overhead that complicates masked-token pretraining. These drawbacks are magnified in fMRI: spatial grids are high resolution and a large fraction of voxels are non-brain background. Hiera shows that many hand-crafted components are unnecessary. A minimal hierarchical ViT by pretext task achieves superior speed/accuracy trade-offs and is naturally compatible with MAE-style pretraining (Ryali et al., 2023). For voxel-level 4-D fMRI, this is particularly attractive: with MAE-style masking, non-informative background can be excluded from the encoder, reducing GPU memory and improving throughput while preserving fine-grained brain signals.

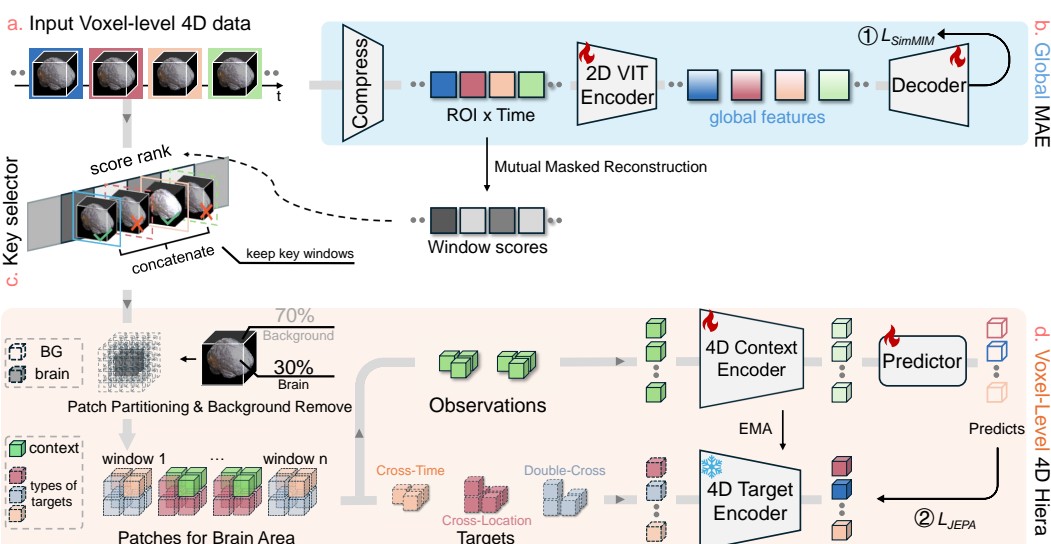

Figure 3: **SLIM-Brain Pipeline.** (a) Atlas-free 4D fMRI input at voxel resolution. (b) A lightweight ViT processes the full recording to produce robust global features. (c) Using the same masking mechanism, a cross-window masked-reconstruction score ranks temporal windows and selects informative segments. (d) Selected windows are routed to a voxel-level 4D Hiera encoder to extract fine-grained representations without any predefined atlas.

## 3 METHODS

SLIM-Brain is an atlas-free 4D fMRI encoder that operates directly on voxel-level volumes (Fig. 3a). We first summarize full-length recordings with a lightweight ViT trained using masked autoencoding (MAE; SimMIM-style) to obtain robust global features (Fig. 3b). Using the same masking machinery, we compute a *mutual masked reconstruction* score to rank temporal *windows* and select the informative ones (Fig. 3c). The selected windows are then routed to a 4D Hiera encoder, which extracts fine-grained voxel-level representations without any predefined atlas (Fig. 3d). This selective-compute design avoids pushing large non-brain background through the encoder and sidesteps redundant volumetric reconstruction, yielding substantial gains in speed and GPU memory efficiency while preserving voxel-level detail.

## 3.1 GLOBAL FEATURES VIA MASKED AUTOENCODING

Given a 4D fMRI time series $\mathbf{X} \in \mathbb{R}^{H \times W \times D \times T}$, we first partition the spatial volume $(H, W, D)$ into non-overlapping cubic patches of size $(u, u, u)$, yielding $B$ candidate patches. Using a brain mask, we discard any patch whose voxels are all background, resulting in $b$ valid patches. For each valid patch, we average the voxel values within the patch at every time point to obtain a single time series of length $T$. Stacking these patch-wise time series yields a 2D matrix $\mathbf{X}' \in \mathbb{R}^{b \times T}$, where the first dimension $b$ indexes the retained patches and the second dimension $T$ indexes time.

Next, we partition the temporal axis of $\mathbf{X}'$ into non-overlapping patches of length $p$ (zero-padding the tail if needed), producing

$$\mathbf{X}'' \in \mathbb{R}^{b \times M \times p}, \qquad M = \left\lceil \frac{T}{p} \right\rceil.$$

By stacking the $bM$ patches, we form a token matrix $\mathbf{P} \in \mathbb{R}^{(bM) \times p}$.

The global branch follows a masked autoencoding (MAE) scheme (Fig. 3b). We randomly mask a ratio $r$ of tokens and feed the whole masked data $\mathbf{P}_{\text{masked}}$ to a Vision Transformer encoder $\mathcal{E}$:

$$\mathbf{z} = \mathcal{E}(\mathbf{P}_{\text{masked}}) \in \mathbb{R}^{bM \times C},$$

where $\mathbf{z}$ is the representation. A lightweight reconstruction head $\mathcal{R}$ predicts the original data,

$$\widehat{\mathbf{P}} = \mathcal{R}(\mathbf{z}) \in \mathbb{R}^{bM \times p},$$

and training uses a SimMIM-style objective on all positions:

$$\mathcal{L}_{\text{SimMIM}} = \frac{1}{bMp} \left\| \widehat{\mathbf{P}} - \mathbf{P} \right\|_2^2.$$

This global MAE path yields an integrated representation of the full-length sequence, preserving long-range dynamics even when only a small subset of top-ranked windows is subsequently processed by the heavy 4D encoder.

## 3.2 TOP-$k$ SELECTOR VIA MUTUAL MASKED RECONSTRUCTION.

We then rank temporal windows using the pretrained frozen global MAE as a context learner which was pretrained on the same dataset as the voxel-level 4D Hiera encoder. (Fig. 3c). Given $M$ non-overlapping temporal windows, the MAE captures window-level structure and provides a principled signal for assessing each window's contribution to the whole.

For a candidate window $m \in \{1, \dots, M\}$, we keep only window $m$ and mask the remaining $M-1$ windows which window $\zeta$, then run the frozen MAE to reconstruct the masked ones. Let $Y_j$ denote the ground-truth content of window $j$ and $\hat{Y}_j^{(m)}$ the reconstruction when only window $m$ is provided. We define the *mutual masked reconstruction* score as the negative reconstruction error averaged over all masked windows:

$$s_m = -\frac{1}{M-1} \sum_{j \neq m} \text{MSE}\left(\hat{Y}_j^{(m)}, Y_j\right),$$

so that higher $s_m$ indicates stronger global representativeness (i.e., patch $m$ better supports reconstructing the rest of the sequence). We then select the top-$k$ windows where each window consists of $\zeta$ consecutive frames,

$$\mathcal{T} = \text{Topk}\left(\{s_m\}_{m=1}^M, k\right),$$

and feed their corresponding 4D sub-volumes to the voxel-level 4D encoder to extract fine-grained spatiotemporal representations for downstream modeling.

## 3.3 VOXEL-LEVEL FEATURES WITH A 4D HIERA ENCODER

We adopt a dual-branch Hiera encoder in the voxel-level path (Fig. 3d). Hiera is a hierarchical Transformer in which local self-attention is restricted to mask units, replacing the Swin-style shifted windows. This design accommodates irregular inputs by operating at the unit level, allowing background units and any units designated for masking to be pruned outright. For each of the top-$k$

windows from a continuous $T$-frame fMRI sequence, we partition the volume into a regular grid of mask units of size $u$, remove units that are entirely background, and retain a sparse set of foreground units $\mathcal{U}$. Within each retained unit, we apply a $n \times n \times n$ patch-merge operation, and we add spatial and temporal positional encodings to preserve the full 4D structure.

The path is trained with a JEPA objective and comprises a context encoder $\mathrm{Enc}_c$ and a target encoder $\mathrm{Enc}_t$. At each iteration, the context view $\mathcal{C} \subset \mathcal{U}$ is formed by sampling a spatiotemporally contiguous block covering approximately $40\%$ of units. The target view $\mathcal{M}$ is sampled to be non-overlapping with the context and instantiated in three ways: (i) different spatial units at the same time, (ii) the same spatial units at different times, and (iii) a spatiotemporal combination of (i) and (ii). Crucially, only the units that belong to a given view are fed to the corresponding encoder—there is no need to feed all patches and then mask them as in Swin-style tiling.

We encode the two views with a dual-branch Hiera:

$$H_c \;=\; \mathrm{Enc}_c\big(X_{\mathcal{C}}\big), \qquad H_t \;=\; \mathrm{Enc}_t\big(X_{\mathcal{M}}\big) \text{ (stop-grad)}.$$

A ViT-style predictor $\mathrm{Pred}$ maps context features to the target space at the masked indices, producing $\widehat{H}_t \;=\; \mathrm{Pred}(H_c; \mathcal{M}) \in \mathbb{R}^{|\mathcal{M}| \times C}$. We optimize a JEPA-style regression with Smooth-$\ell_1$:

$$\mathcal{L}_{\mathrm{JEPA}} \;=\; \frac{1}{|\mathcal{M}|} \sum_{p \in \mathcal{M}} \mathrm{SmoothL1}\big(\widehat{H}_t[p],\, H_t[p]\big).$$

The target branch is an exponential–moving–average (EMA) copy of the context branch:

$$\theta_t \leftarrow \tau\,\theta_t + (1 - \tau)\,\theta_c, \quad \tau \in [0, 1),$$

where $\theta_c$ and $\theta_t$ are the parameters of $\mathrm{Enc}_c$ and $\mathrm{Enc}_t$.

### 3.4 INFERENCE PIPELINE.

Given a full-length fMRI sequence, the global MAE branch yields a compact global descriptor $\mathbf{z} \in \mathbb{R}^C$ and ranks temporal windows via mutual masked reconstruction. The top-$k$ windows are concatenated and passed to the 4D Hiera encoder, whose outputs are subsequently pooled to yield a fine-grained local descriptor $g_{\text{top-k}} \in \mathbb{R}^{C_{\text{mid}}}$. We then apply average pooling followed by simple MLP layers for downstream tasks. For linear probing, the backbone is frozen and only the MLP is trained, whereas for fine-tuning, the entire model is updated end-to-end.

## 4 EXPERIMENT

### 4.1 DATASET

We leverage self-supervised pretraining on four public neuroimaging datasets: a single HCP session (Van Essen et al., 2013), CHCP (Ge et al., 2023), and two releases from the Amsterdam Open MRI Collection (AOMIC)—PIOP1 and PIOP2 (Snoek et al., 2021); the Adolescent Brain Cognitive Development (ABCD) Study Casey et al. (2018). We use 70% of the data (4129 sessions) for training, with the remainder reserved for validation (10%) and testing (20%). External validation spans four datasets: the Autism Brain Imaging Data Exchange (ABIDE) (Di Martino et al., 2014); the ADHD-200 Sample (ADHD) (consortium, 2012); Alzheimer's Disease Neuroimaging Initiative (ADNI) Jack Jr et al. (2008); Parkinson's Progression Markers Initiative (PPMI) Marek et al. (2011). See Appendix A.1 and A.2 for preprocessing and dataset details.

### 4.2 FINE-TUNING RESULTS

We fine-tune our model and evaluate it on out-of-distribution datasets, as summarized in Table 1 and Appendix A.3. We benchmark against seven representative fMRI models—BrainNetCNN (Kawahara et al., 2017), BrainGNN Li et al. (2021), BrainLM (Caro et al., 2023), BrainMass (Yang et al., 2024), Brain-JEPA (Dong et al., 2024), NeuroSTORM Li et al. (2025) and SwiFT Kim et al. (2023)—using the authors' released codes or checkpoints (pretrained on 10k–64k samples). Note that BrainMass includes these datasets in its pre-training corpus, whereas our method utilizes them

Table 1: **Performance on external tasks.** We report the fine-tuning disease classification accuracy (%) for **ADHD**, **ADNI** and **PPMI**; age classification and regression (ACC% and MSE) for **ABIDE**. "Samples (K)" denotes the number of pre-training sessions (in thousands). The symbol * indicates statistical significance over all baselines with $p < 0.05$. Data is presented as mean $\pm$ standard deviation.

| Model | Samples (K) | ADHD-200 | ADNI (MCI) | PPMI | ABIDE age | |
|---|---|---|---|---|---|---|
| | | ACC% ↑ | ACC% ↑ | ACC% ↑ | ACC% ↑ | MSE ↓ |
| BrainNetCNN | - | $54.46 \pm 2.47$ | $59.74 \pm 4.50$ | $64.24 \pm 2.41$ | $41.52 \pm 3.49$ | $0.7025 \pm 0.028$ |
| BrainGNN | - | $55.87 \pm 3.88$ | $63.20 \pm 7.38$ | $55.56 \pm 4.81$ | $33.17 \pm 4.46$ | $0.9338 \pm 0.000$ |
| BrainLM | 42 | $57.86 \pm 0.00$ | $61.41 \pm 0.09$ | $66.67 \pm 1.04$ | $39.24 \pm 4.36$ | $0.8700 \pm 0.059$ |
| BrainMass | 65 | $60.78 \pm 0.49$ | $62.39 \pm 0.60$ | $63.51 \pm 0.49$ | $48.19 \pm 1.64$ | $0.5129 \pm 0.042$ |
| Brain-JEPA | 32 | $59.74 \pm 0.23$ | $64.53 \pm 0.60$ | $64.57 \pm 1.79$ | $34.00 \pm 2.14$ | $0.2704 \pm 0.044$ |
| SwiFT | 10 | $60.81 \pm 2.38$ | $64.45 \pm 1.69$ | $58.10 \pm 0.00$ | $62.22 \pm 0.55$ | $0.4137 \pm 0.033$ |
| NeuroSTORM | 58 | $62.35 \pm 0.90$ | $66.67 \pm 1.06$ | $69.12 \pm 0.99$ | $38.64 \pm 2.14$ | $0.5890 \pm 0.066$ |
| SLIM-Brain | 4 | $\textbf{63.53} \pm \textbf{0.53*}$ | $\textbf{69.12} \pm \textbf{1.38*}$ | $\textbf{70.40} \pm \textbf{0.59}$ | $\textbf{64.41} \pm \textbf{0.57*}$ | $\textbf{0.2175} \pm \textbf{0.019*}$ |

strictly for fine-tuning. We adopted the medium-sized model (45M parameters) trained on 4,129 sessions as the default configuration for the experiments. Across all the OOD evaluations, our model consistently outperforms the baselines, indicating that it learns domain-invariant representations that transfer to unseen datasets and tasks with minimal adaptation. Task description (A.2), statistical method (A.3), internal experiment results (A.4), linear probing results results (A.5), configuration details for ours (A.6) and baselines (A.7) are provided in the Appendix.

Our approach is markedly more compute- and data-efficient. Whereas prior work reports training Brain-JEPA for 300 epochs on 4×A100 GPUs, BrainMass for 2000 epochs on 8×V100 GPUs ($\sim$ 150 hours) and NeuroSTORM for 30 epochs on 4×A6000 GPUs (48 GB) ($\sim$ 13 days) our model (4K sessions) trains for 40 epochs on 1× A100 GPUs (80 GB) in about 20 hours, including I/O and computing. On the data side, contemporary fMRI foundation models are typically pretrained at massive scale, for example, BrainMass is pretrained on 26 datasets (including HCP, UKB, and ABIDE) totaling 64,584 subjects, whereas our pretraining uses 4,129 sessions.

## 4.3 ABLATION STUDY

**Ablation on key frames selection strategy.** To assess the effect of key-frame selection on downstream performance, we evaluate four strategies for ADHD disease classification (Table 2) by *linear probing*. (1) *Top-k (ours)*, (2) *Temporal variance*, (3) *Uniform sampling*, (4) *Random sampling*. Strategies are detailed in Appendix A.8. The temporal variance–based method achieves competitive performance, suggesting that selecting nonredundant frames is beneficial, although its redundancy measure is less accurate than Top-$k$ (Welch's t-test, $p = 0.0153$). Uniform and random sampling perform substantially worse, indicating that naive temporal downsampling fails to preserve discriminative disease-related dynamics (Welch's t-test, $p = 0.0223$). Moreover, the improvement of Top-$k$ over random sampling is statistically significant (Welch's t-test, $p = 0.0004$), confirming that the gain is not attributable to noise.

Table 2: **Table B: Ablation study on key frame selection strategies (ADHD).** We report Mean and STD over 3 independent runs. The symbol * indicates statistical significance ($p < 0.05$).

| Strategy | ACC% (Mean $\pm$ STD) | F1% (Mean $\pm$ STD) | Description |
|---|---|---|---|
| Random | $56.0 \pm 0.6$ | $56.1 \pm 0.7$ | No heuristic |
| Uniform Sampling | $56.7 \pm 1.4$ | $50.9 \pm 9.9$ | Fixed intervals |
| Temporal Variance | $57.2 \pm 1.2$ | $56.4 \pm 2.2$ | Correlation-based |
| **Top-K (Ours)** | $\textbf{61.1} \pm \textbf{0.5*}$ | $\textbf{61.0} \pm \textbf{0.7*}$ | **Learnable selector** |

**Scaling study** We evaluated the scalability of SLIM-Brain by comparing its performance by ADNI (**AD** vs. CN) across varying pre-training dataset sizes (606, 1,037, 4,129) and model parameter counts (22 M, 45 M, 87 M). As illustrated in Fig. 4, our approach demonstrates strong scaling capabilities: accuracy improved from 73.72% to 80.09% as the dataset size increased, and from 77.32% to 85.50% as the model size increased. Crucially, we observed no signs of performance saturation even at the largest scale, suggesting that SLIM-Brain follows neural scaling laws similar to those in natural language processing. This indicates that further increasing the pre-training corpus or model capacity could yield continued gains in diagnostic precision. To ensure robustness, all experiments were conducted with three independent runs. Detailed performance metrics, including the means and variances of accuracy and F1-scores, are provided in the Appendix A.9.

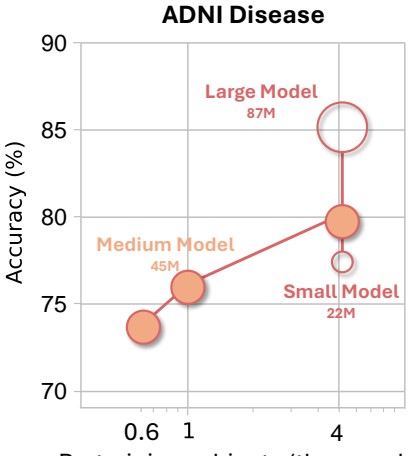

Figure 4: **Scaling study.** Performance on AD vs. CN at varying amounts of pre-training data and model parameter sizes..

**Top-$k$ windows outperform random selection.** To avoid ingesting full-length fMRI sequences, we use the global MAE to obtain a coarse descriptor and to score temporal windows; only the top-ranked windows are fed to the 4D encoder (Table 3). We use five consecutive frames to form one window, and multiple such windows are grouped into a set as the input to the voxel-level model. With a per-frame storage layout, we can load arbitrary time indices on demand, cutting $\sim 80\%$ of 4D data I/O and compute when selecting the top 20% of windows, comparing sliding windows average method (Kim et al., 2023). We compare *Top-$k$* selection (mutual masked reconstruction) against Random selection under different frame budgets on HCP sex classification with 1K-size pre-trainging model. Top-$k$ consistently achieves higher accuracy and F1 without full-sequence 4D fMRI reads.

| Set size | 5 frames (k=1) | 20 frames (k=4) | 40 frames (k=8) |
|---|---|---|---|
| Random | 83.9 / 83.7 | 84.5 / 84.1 | 86.0 / 85.9 |
| Top-K | **84.5 / 84.2** | **86.7 / 86.6** | **87.7 / 87.6** |

Table 3: Ablation on window selection (**HCP** sex). Classification accuracy (%) / F1-score (%) at different frame budgets.

**Ablations on architectural choices.** We assess the impact of our design decisions on memory and accuracy by 1K-size pretraining model with HCP dataset in Table. 4. First, employing a 4D Hiera encoder with unit-wise attention excludes background and masked units from the forward pass, substantially reducing the memory footprint. Because fMRI signals are not directly interpretable like natural images, we prioritize representation quality over pixel-space fidelity and adopt a JEPA objective, dispensing with a heavy reconstruction decoder. With 200-frame inputs and Top-$k$ selection, our design reduces peak GPU memory from $\sim 8$ GB to $\sim 2.4$ GB *while improving accuracy*. Meanwhile, although Hiera–MAE and Hiera–JEPA have comparable per-sample GPU memory footprints, the JEPA variant achieves substantially higher throughput. By contrast, Kim et al. (2023) processes ten 20-frame windows sequentially and averages their predictions; each 20-frame chunk consumes $\approx 3.2$ GB of GPU memory, so handling all windows inflates the per-sample memory footprint. Likewise, Kwon et al. (2024) employ a Swin-UNet–based model for task prediction; although it is not a foundation model and does not reconstruct to the raw signal space, it still requires $\sim 40$ GB of GPU memory to process 30 volumes with a batch size of four. Details are in Appendix A.10

### 4.4 MODEL INTERPRETATION

We examine whether SLIM-Brain's voxel-level representations are neurobiologically meaningful. First, we use *Neurosynth*—a large-scale, automated meta-analytic platform that aggregates findings from thousands of fMRI studies (Yarkoni et al., 2011)—to obtain disease-associated meta-analytic

Table 4: Ablation on structure choices on 1K pre-training models. *Memory* denotes GPU demand per sample (200 frames; GB).

| Model | HCP Sex | | HCP Fingerprint | | ABIDE Age | | memory ↓ |
|---|---|---|---|---|---|---|---|
| | ACC ↑ | F1 ↑ | ACC ↑ | F1 ↑ | ACC ↑ | F1 ↑ | |
| Swin-SIM | 90.8 | 90.7 | 38.2 | 27.6 | 59.3 | 59.5 | 8.0 |
| Swin-JEPA | 87.3 | 87.3 | 84.0 | 79.8 | **62.1** | **63.1** | 4.0 |
| Hiera-MAE | **91.3** | **91.3** | 90.0 | 87.4 | 52.6 | 52.6 | 2.4 |
| Hiera-JEPA | 91.1 | 91.1 | **98.5** | **98.1** | 59.6 | 58.3 | **2.3** |

maps (e.g., ADHD). Next, we derive fine-grained attention maps from the 4D Hiera encoder by projecting token-level attention back to voxel space. (Implementation details are provided in the Appendix A.11.) Integrated-gradients–derived key regions from SLIM-Brain substantially overlap with Neurosynth meta-analytic distributions (Fig. 5). The highlighted regions include dorsolateral prefrontal cortex and anterior cingulate (executive control), inferior parietal and precuneus/posterior cingulate (default mode network), as well as striatal areas, consistent with ADHD-related fronto-striatal and fronto-parietal dysfunctions. These observations indicate that the learned features are not only predictive but also align with established neurobiological patterns, offering interpretable, voxel-level evidence directly from raw fMRI inputs.

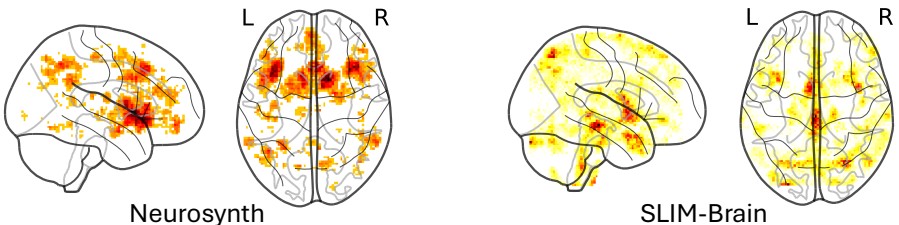

Figure 5: **Fine-grained attention map shows key brain area.** Left: ADHD results from Neurosynth. Right: Integrated-gradients–derived key regions from SLIM-Brain.

## 5 DISCUSSION

Recent fMRI foundation models have demonstrated strong performance on diverse downstream tasks, but most simplify inputs via *atlas-based parcellation*: the brain is first divided into template-defined regions and BOLD signals are averaged within each ROI. While this streamlines computation, it can introduce parcellation bias and suppress voxel-level transients that matter for fine-grained cognitive decoding. At the other extreme, feeding full 4D volumes avoids parcellation but is computationally demanding and, when paired with global mean pooling, still risks washing out short-lived patterns. SLIM-Brain sidesteps both spatial parcellation and coarse temporal pooling with a two-step coarse-to-fine design. Temporally, a top-$k$ window selector prevents full sequences from being loaded into GPU memory. Spatially, the Hiera–JEPA encoder prunes non-brain background patches and routes each branch to its own patch subset (context vs. target) without materializing the full token grid, yielding fine-grained voxel-level features while using only $\approx 30\%$ of the GPU memory required by Swin-based models. Empirically, with pretraining on ∼4k subjects, SLIM-Brain surpasses recent ROI/parcellation-based baselines trained on tens of thousands of subjects across multiple tasks, suggesting that voxel-level modeling captures discriminative information that atlas averaging can obscure. At the same time, the lightweight memory footprint broadens practical applicability (e.g., commodity GPUs) and makes larger-scale pretraining more tractable, pointing toward scalable, atlas-free voxel-level foundation models for fMRI.

**Limitations and future work.** First, although our design substantially reduces GPU memory, full-resolution 4D fMRI still imposes a nontrivial I/O burden. In practice, high-throughput storage (e.g., SSDs) and I/O-aware layouts are important. Second, our Top-$k$ selection uses *mutual masked reconstruction*, which favors windows that best represent the remainder of the sequence. This criterion suits resting-state data but may down-weight task-evoked segments precisely because they are unique; hybrid scores that blend representativeness and event uniqueness are a natural extension.

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

# A APPENDIX

## A.1 PREPROCESSING

HCP and CHCP data were preprocessed using the HCP minimal preprocessing pipeline (Glasser et al., 2013). ABCD data were processed using the ABCD-HCP pipeline. PIOP1, PIOP2 and ISYB datasets were preprocessed using fMRIPrep (Esteban et al., 2019). For ABIDE and ADHD, we adopted preprocessed datasets provided by the Preprocessed Connectomes Project (PCP) (Craddock et al., 2013; Bellec et al., 2017). To ensure compatibility across datasets, we first resampled all images to a uniform spatial resolution of 2 mm isotropic using cubic B-spline interpolation. For datasets with lower temporal resolution, we further interpolated the time series to a uniform sampling rate of 0.72 s TR, also using cubic B-spline interpolation. Notably, HCP, CHCP, and ABCD natively share nearly identical temporal resolution and already meet the 2 mm spatial resolution requirement after preprocessing.

## A.2 DATASET AND DOWNSTREAM TASKS

**Human Connectome Project (HCP).** We utilize the HCP dataset as our primary benchmark for pre-training and initial evaluation, comprising 1,011 subjects. This cohort represents a young adult population with ages ranging from 22 to over 36 years. Specifically, the age distribution includes distinct groups: 22–25 ($N = 218$), 26–30 ($N = 437$), 31–35 ($N = 346$), and 36+ ($N = 10$). The dataset maintains a balanced sex ratio with 469 males and 542 females, providing high-quality, standardized data ideal for establishing baseline model performance. We only use one session per participants Van Essen et al. (2013). **HCP Fingerprint** Fingerprint serves as a subject-identification benchmark. We randomly chose 100 subjects as distinct classes. Three different sessions from the same subjects are assigned to train, validation, and test splits, ensuring that the task evaluates cross-session fingerprint consistency rather than session-specific artifacts.

**Amsterdam Open MRI Collection (AOMIC).** We include two independent datasets from the AOMIC project for pretraining. PIOP1 consists of 203 subjects with a mean age of $22.208 \pm 1.795$ years (86 males, 117 females). PIOP2 comprises 222 subjects with a similar mean age of $21.946 \pm 1.783$ years (95 males, 127 females).

**Chinese Human Connectome Project (CHCP).** To enhance the demographic diversity of our pre-training corpus, we incorporate the CHCP dataset, which consists of 304 subjects (145 males,

159 females).The CHCP cohort presents a significantly broader age distribution ($29.326 \pm 14.860$ years).

**Adolescent Brain Cognitive Development (ABCD).** To evaluate model generalization on a larger scale and a distinct demographic, we incorporate the ABCD dataset, which includes 2,398 subjects Casey et al. (2018). In contrast to HCP, this cohort focuses on early brain development, with a tightly clustered mean age of $9.925 \pm 0.627$ years. And the dataset features a highly balanced sex distribution (1,226 males and 1,172 females).

**ADHD Disease** ADHD-200 is an open multi-site neuroimaging resource with resting-state fMRI (rs-fMRI) across multiple contributing centers consortium (2012). For this task we merge all ADHD subtypes into a single ADHD label and it contains 938 participants (ADHD 356 vs. control 582).

**Autism Brain Imaging Data Exchange (ABIDE)** The Autism Brain Imaging Data Exchange is a multi-site, open-access initiative that aggregates structural and resting-state fMRI scans - alongside rich phenotypic data - from individuals with autism spectrum disorder (ASD) and matched typically developing controls to accelerate reproducible neuroimaging research Di Martino et al. (2014). It contains 871 participants (mean age 16.9, std 7.5). We perform regression on age and scale both the labels and results to the range [0, 1] using StandardScaler. Additionally, we divide the age variable into four balanced groups for the classification task.

**Alzheimer's Disease Neuroimaging Initiative (ADNI)** The Alzheimer's Disease Neuroimaging Initiative is a large-scale, longitudinal, multi-center project initiated in 2004, designed to collect and publicly share clinical data from cognitively normal older adults, individuals with mild cognitive impairment (MCI) and patients with Alzheimer's disease (AD) Jack Jr et al. (2008). For our task, we separated disorders into two classification with healthy controls.

**Parkinson's Progression Markers Initiative (PPMI)** The Parkinson's Progression Markers Initiative is an ongoing, international, longitudinal observational study launched in 2010 by The Michael J. Fox Foundation to identify and validate biomarkers of Parkinson's disease (PD) risk, onset, and progression. We combined two different diseases and healthy controls from the PPMI dataset to create a three-class classification task Marek et al. (2011). This cohort consists of 474 subjects (272 males, 202 females; mean age 66.44). We utilize the data for a three-class **disease classification** task: Prodromal (303), PD (142), and Control (29).

## A.3 STATISTICAL ANALYSIS OF EXTERNAL TASKS.

To ensure the statistical reliability of our results, we conducted 3 independent runs for all Out-of-Distribution (OOD) benchmarks.And we calculates the p-values for each pair of models based on their respective accuracy values across three runs. For each model pair, the null hypothesis is that the means of the two models' accuracies are equal, and the test assesses whether there is sufficient evidence to reject this hypothesis. The p-values are stored in a matrix and the most statistically significant comparison is identified by finding the pair with the smallest p-value. The result indicates that our model statistically significant improvements ($p < 0.05$) across these diverse tasks.

## A.4 INTERNAL EXPERIMENTS

We fine-tune our model and evaluate it on the held-out 20% split of the HCP dataset (sex classification and subject identification/fingerprinting) as summarized in Table 5. We benchmark against two representative fMRI foundation models, BrainMass (Yang et al., 2024), and Brain-JEPA (Dong

et al., 2024)—using the authors' released checkpoints (pretrained on 32k–64k samples). For a controlled comparison under matched data budgets, we additionally retrain each baseline on the same total number of sessions as ours and report results for their best validation checkpoints (Table 5 and Fig. 1).

Table 5: Internal task. Fine-tuning accuracy (%) and F1-score (%) on **HCP** Sex and fingerprint. "Samples (K)" is the number of pretraining sessions (in thousands). "LP" means results from linear probing.

| Model | Samples (K) | Sex | | Fingerprint | |
|---|---|---|---|---|---|
| | | Acc ↑ | F1 ↑ | Acc ↑ | F1 ↑ |
| BrainLM | 1 | 62.4 | 61.6 | 47.0 | 41.8 |
| Brain-JEPA | 1 | 54.0 | 35.0 | 1.0 | 0.0 |
| BrainMass | 1 | 78.2 | 78.0 | 43.0 | 36.0 |
| BrainLM | 42 | 74.4 | 77.7 | 51.0 | 41.8 |
| Brain-JEPA | 32 | 87.1 | 85.4 | 57.0 | 48.9 |
| BrainMass | 65 | 77.2 | 77.2 | 86.0 | 82.5 |
| SLIM-Brain (LP) | 0.6 | 90.1 | 90.0 | 89.0 | 87.2 |
| SLIM-Brain (LP) | 1 | 90.6 | 90.5 | 98.5 | **98.4** |
| SLIM-Brain | 1 | **91.1** | **91.1** | **98.5** | 98.1 |

## A.5 LINEAR PROBING RESULTS

To assess the intrinsic quality of the learned representations—independent of task-specific adaptation—we conduct *linear probing* (LP). LP freezes the pretrained encoder and trains a lightweight linear classifier on top, providing a direct measure of how well the latent features capture task-relevant information and their linear separability He et al. (2022). We evaluate LP on two downstream tasks, focusing on ADNI diagnosis (Table 6). Despite pretraining on a small subset of data, our linear probes transfer robustly to out-of-distribution (OOD) settings, outperforming baseline encoders and, in some cases, even their fine-tuned counterparts. At the same time, compared to MAE-based models, our method shows less performance degradation when using linear probing (2.5% vs. 5.1%).

Table 6: **Linear probing** Comparison of linear probing performance on ADNI (MCI). We report Accuracy and F1-Score over 3 independent runs and computed mean and standard deviation.

| Model | Finetune Acc | Finetune F1 | Linprobe Acc | Linprobe F1 |
|---|---|---|---|---|
| NeuroSTORM (MAE-based) | $66.67 \pm 0.60$ | $65.58 \pm 1.74$ | $61.54 \pm 0.00$ | $46.89 \pm 0.00$ |
| SLIM-Brain (Ours) | $69.12 \pm 1.38$ | $68.96 \pm 0.26$ | $66.66 \pm 1.40$ | $64.52 \pm 1.16$ |

## A.6 EXPERIMENTAL SETTINGS

Unless otherwise stated, we use 4D fMRI blocks of size $96 \times 96 \times 96 \times 40$ $(H, W, D, T)$ ; train for 8 epochs with a global batch size of 32 on NVIDIA L40 GPUs; use Adam with learning rate $1 \times 10^{-3}$ for pretraining; apply an MAE masking ratio $r = 0.75$; spatially downsample to a $12 \times 12 \times 12$ lattice $(H' = W' = D' = 12)$; To construct the 2D global data, we partition the volume into non-overlapping cubic patches with size $u = 8$, resulting in $b = 716$ foreground blocks.; take a clip length $T = 200$; set window length $p = 5$ giving $M = \lceil T/p \rceil = 40$ windows; keep the top-$k$ windows with $k = 8$ (i.e., 40 frames when $p = 5$); use mask-unit size $u = 24$ voxels per side; and a patch-merge kernel $n = 6$ $(6 \times 6 \times 6)$. For JEPA, the context set covers $40\%$ of foreground units per iteration (non-overlapping target set is the remainder).

To ensure a fair comparison across baseline models, we conduct experiments with each baseline using three random seeds. We begin by performing a grid search over key hyperparameters, while adhering to the default settings specified in the original implementations when available. For hyperparameters not explicitly mentioned in our introduction of the baseline models, we follow the

default configurations provided. Given that accuracy can be misleading on imbalanced datasets, we report weighted F1 scores as the primary evaluation metric. Detailed training and fine-tuning logs are provided at `https://anonymous.4open.science/r/SLIM-Brain-9C51`.

## A.7 DETAILS OF BASELINE MODELS

In this section, we introduce our baselines models. First four foundational models for brain activity prediction: BrainLM, BrainMASS, Brain-JEPA and NeuroSTORM, each of which uses distinct parcellation schemes and is trained on large-scale, multi-site fMRI datasets. We also fine-tuned our downstream tasks using other end-to-end models: BrainNetCNN, BrainGNN, SwiFT.

**BrainNetCNN** is a convolutional neural network specifically designed for analyzing connectome-structured data. Unlike traditional models that process images or graphs, BrainNetCNN employs specialized convolutional filters—edge-to-edge, edge-to-node, and node-to-graph—that directly model pairwise connectivity patterns. In our experiments, the model takes as input a functional connectivity (FC) matrix, which is computed using Pearson correlation coefficients between regional fMRI time series. We set the hyperparameters as follows: batch size of 64, learning rate of 0.01, and weight decay of 0.05.

**BrainGNN** is a graph neural network architecture designed to analyze brain connectomes at the population level by learning interpretable node- and region-specific biomarkers. For each subject, it construct a graph whose nodes correspond to brain regions and whose edges encode statistical dependencies between regional fMRI time series. The functional connectivity (FC) matrix is used as the node-level feature representation, and subject-level labels provide supervision for learning anatomically meaningful subgraphs and node embeddings. Partial correlations between fMRI time series are computed and assigned as edge attributes, enabling the model to capture conditional dependencies between brain regions rather than only marginal pairwise correlations. During training, we use batch size of 64, learning rate of 0.001, and weight decay of 0.5. It is important to note that, unlike the multiple regularization terms used in classification tasks, such as entropy regularization and consistency regularization, we only utilize the Mean Squared Error (MSE) loss for the age regression task to minimize adaptations. However, this approach has led to unstable learning in the BrainGNN model, resulting in its failure to converge on this task, even after extensive hyperparameter tuning.

**BrainLM** is the first fMRI foundation model specifically designed to capture the spatiotemporal dynamics of brain activity through a Transformer-based masked autoencoder architecture. It employs the AAL-424 atlas for the parcellation of brain regions, transforming fMRI recordings into a 424-dimensional set of regions of interest (ROIs). The model is trained on an extensive dataset consisting of 6,700 hours of fMRI data derived from 77,298 recordings across the UK Biobank and the Human Connectome Project. For downstream fine-tuning, we choose the following hyperparameters: batch size of 64, learning rate of 0.00001, and training duration of 30 epochs.

**BrainMASS** utilizes the Schaefer 100-region atlas for the parcellation of brain networks. It was trained on 26 datasets, encompassing a total of 64,584 subjects, including data from the UK Biobank (UKB), Human Connectome Project (HCP), and ADHD-200, etc. The model is designed to learn representations of functional brain networks by integrating both masked ROI modeling and latent representation alignment techniques, thereby enhancing its diagnostic performance. The following hyperparameters were selected for downstream fine-tune: a batch size of 64, a learning rate of 0.0002, and a training duration of 100 epochs.

**Brain-JEPA** employs an advanced parcellation strategy that combines the Schaefer-400 cortical regions with the Tian-Scale III subcortical regions, resulting in a total of 450 regions of interest (ROIs). The model is trained on 80 % of the UK Biobank (UKB) data and utilizes a joint-embedding predictive architecture. This architecture incorporates both spatial and temporal masking techniques to capture dynamic patterns of brain activity. The model is trained with the following hyperparameters: a batch size of 16, a learning rate of $4 \times 10^{-4}$, over a total of 50 epochs.

**NeuroSTORM** establishes a general-purpose neuroimaging foundation model that learns directly from raw 4D fMRI volumes, avoiding the information loss associated with projecting data

onto pre-defined atlases or connectomes. It employs a Shifted-Window Mamba (SWM) backbone. The model is pre-trained on a large-scale fMRI data from over 50,000 subjects—spanning the UK Biobank, ABCD, and HCP datasets. The model utilizes a Spatiotemporal Redundancy Dropout (STRD) module to mitigate data redundancy while capturing long-range dependencies.

SwiFT (Swin 4D fMRI Transformer) addresses the challenge of modeling high-dimensional spatiotemporal brain dynamics by learning directly from raw 4D fMRI volumes, thereby avoiding the information loss introduced by hand-crafted features. Through its 4D windowed attention mechanism and computationally efficient architecture, SwiFT outperforms recent state-of-the-art models on large-scale datasets. Moreover, contrastive self-supervised pretraining further enhances its downstream performance, highlighting SwiFT's effectiveness as an end-to-end framework for functional brain imaging. In our experiments, we fine-tune SwiFT with a batch size of 16, a learning rate of $1 \times 10^{-6}$, a weight decay of 0.01, and a training duration of 9 epochs. Due to SwiFT's substantial I/O demands, we run each downstream task only once.

### A.8  DETAILS OF KEY FRAME SELECTION

To assess the effect of key-frame selection on downstream performance, we evaluate four strategies for ADHD disease classification (Table 2). (1) *Top-k*. We employ the global MAE to estimate the informativeness of each window, assigning higher scores to segments with larger reconstruction errors, which indicate lower redundancy and richer signal content. (2) *Temporal variance*. Instead of relying on model predictions, this strategy computes a frame–frame correlation matrix which measures its correlation with all other frames. Frames with the lowest mean correlation are selected, capturing temporally unique BOLD patterns that contribute nonredundant information. (3) *Uniform sampling*. The uniform strategy samples one frame at fixed temporal intervals, providing evenly spaced coverage across the sequence without accounting for signal variation. (4) *Random sampling*. This strategy randomly selects $k$ frames from the given 4D fMRI sequence without applying any selection heuristic. We observe that Top-$k$ consistently achieves the highest classification accuracy compared with other strategies. The temporal variance–based method achieves competitive performance, suggesting that selecting nonredundant frames is beneficial, although its redundancy measure is less accurate than Top-$k$ (Welch's t-test, $p = 0.0153$). Uniform and random sampling perform substantially worse, indicating that naive temporal downsampling fails to preserve discriminative disease-related dynamics (Welch's t-test, $p = 0.0223$). Moreover, the improvement of Top-$k$ over random sampling is statistically significant (Welch's t-test, $p = 0.0004$), confirming that the gain is not attributable to noise.

### A.9  DETAILS OF SCALING STUDY

We scale our model across both model size and data size. For data scaling, we pretrained a smaller model using only the HCP dataset (606 sessions). For medium-scale pre-train, we utilized HCP, CHCP, and AOMIC datasets (1037 sessions), while for large-scale pre-train, we expanded the data to include the aforementioned datasets along with the ABCD dataset (4129 sessions). For model scaling, we experimented with three different model sizes (22 M, 45 M, 8 7M), adjusting the embedding dimension from 32 to 96 and the depth from 16 to 24.

Following pretraining, we evaluated the performance of the models on the ADNI (AD vs. CN). Each model was run three times, each with different random seeds, while maintaining fixed hyperparameters that were chosen from one seed. The results are shown in Fig. 4, table 7 and table 8.

Table 7: Data scaling on ADNI-AD

| Data Size | Samples(k) | Acc ↑ | F1 ↑ |
|---|---|---|---|
| Small | 0.6 | $73.72 \pm 0.22$ | $72.11 \pm 1.93$ |
| Medium | 1 | $75.93 \pm 2.10$ | $75.69 \pm 0.98$ |
| Large | 4 | $80.09 \pm 0.87$ | $79.62 \pm 0.98$ |

Table 8: Model scaling on ADNI-AD

| Model Size | Pararm (M) | Acc ↑ | F1 ↑ |
|---|---|---|---|
| Small | 22 | $77.32 \pm 1.65$ | $77.12 \pm 1.11$ |
| Medium | 45 | $80.09 \pm 0.87$ | $79.62 \pm 0.98$ |
| Large | 87 | $85.50 \pm 1.16$ | $85.37 \pm 1.11$ |

A.10 DETAILED IMPLEMENTATIONS OF OUR ABLATION STRUCTURES

Beyond Hiera-JEPA, we also evaluated three 4D self-supervised models—Hiera-MAE, Swin-SIM, and Swin-JEPA—whose architectures are detailed below.

**Hiera-MAE** We adapt Hiera-MAE to 4D fMRI by following its hierarchical encoder–decoder design. Stages 1–2 use Mask-Unit Attention, while Stages 3–4 switch to Global Attention. Between stages, Q-pooling progressively downsamples the spatio-temporal tokens to form a compact latent representation. The decoder then upsamples this latent representation to reconstruct the original 4D volume, and the reconstruction error is used as the pretraining loss. Our implementation reuses the official Hiera codebase with systematic modifications to support 4D inputs and volumetric tokenization.

Table 9: Hiera-MAE Pre-training settings (GAS: Gradient accumulation steps; BS: Batch size)

| config | value | config | value |
|---|---|---|---|
| start learning rate | $5 \times 10^{-5}$ | total batch size | 4 GAS × 8 BS |
| learning rate | $1 \times 10^{-4}$ | final learning rate | $1 \times 10^{-6}$ |
| mask ratio | 0.6 | training epochs | 10 |
| weight decay | 0.05 | | |

**Swin-SIM** Complementary to Hiera-MAE, we adopt Swin-SIM with a four-stage hierarchical encoder–decoder: Swin Transformer blocks and patch merging in the encoder yield a compact latent, and a U-Net–style decoder with CNN skip connections restores resolution. We implement this by extending Swin-UNETR and treating time as channels for 4D inputs.

Table 10: Swin-SIM Pre-training settings (same notation as Tab. 9 )

| config | value | config | value |
|---|---|---|---|
| start learning rate | $5 \times 10^{-5}$ | total batch size | 4 GAS × 4 BS |
| final learning rate | $1 \times 10^{-6}$ | training epochs | 14 |
| weight decay | $1 \times 10^{-4}$ | temporal mask ratio | 0.5 |
| spatial mask ratio | 0.6 | | |

**Swin-JEPA** Our Swin-JEPA design combines a four-stage Swin Transformer backbone—Swin blocks within each stage and patch-merging between stages—with a JEPA objective that predicts in latent space rather than pixel space. Concretely, an online encoder processes a context crop and, together with a predictor head, produces a latent that is trained to match the latent from an EMA-updated target encoder applied to a held-out target crop from the same 4D sample. This online-to-target alignment encourages representations that are consistent across spatial/temporal views while avoiding a reconstruction decoder.

Table 11: Swin-JEPA Pre-training settings (same notation as Tab. 9 )

| config | value | config | value |
|---|---|---|---|
| start learning rate | $3 \times 10^{-5}$ | total batch size | 8 GAS $\times$ 8 BS |
| learning rate | $1 \times 10^{-3}$ | final learning rate | $1 \times 10^{-6}$ |
| training epochs | 8 | weight decay | 0.05 |
| pred_mask_R_roi_scale | (0.15, 0.3) | pred_mask_T_roi_scale | (0.2, 0.6) |
| pred_mask_T_roi_scale | (0.2, 0.6) | pred_mask_T_scale | (0.0, 0.4) |

Across all three models we use AdamW as optimizer with a warmup–cosine learning-rate schedule. For data sampling, each sample is partitioned into non-overlapping groups of 200 frames and was uniformly selected a contiguous 40-frame clip (random start index) as the model input.

### A.11 NEUROSYNTH & INTEGRATED GRADIENTS

**Neurosynth**    The uniformity test in Neurosynth is based on a large-scale meta-analysis of thousands of fMRI studies. For a given term (e.g., "ADHD"), Neurosynth collects all studies that reported this term and then computes, at each voxel, the proportion of studies reporting activation in that location. This is statistically compared against the overall base rate of activation across the entire database, yielding a map that highlights voxels more consistently activated in studies mentioning the target term. The resulting map thus reflects consensus activation patterns associated with the disorder.

**Implementation details of disease-associated meta-analytic maps**    We used disease-associated meta-analytic maps from Neurosynth as input references to probe whether SLIM-Brain's voxel-level representations align with established neurobiological findings. Specifically, each meta-analytic map is passed through the pre-trained model, and we apply Integrated Gradients (IG) to compute voxel-wise attributions with respect to the model's predictions.

As a baseline input, we used an empty (all-zero) volume to represent the absence of activation, and interpolated from this baseline to the actual Neurosynth map. The implementation follows the Captum library in PyTorch, which provides efficient IG routines.

### A.12 REPRODUCIBILITY & LLM STATEMENT

Codes and model weights can be found at `https://anonymous.4open.science/r/SLIM-Brain-9C51`. Large Language Models are only used to improve readability and language of the work.

