# OpenReview forum: "SLIM-Brain: A Data- and Training-Efficient Foundation Model for fMRI Data Analysis"
_ICLR.cc/2026/Conference — ICLR 2026 Conference Desk Rejected Submission_

### Official Review · Reviewer_7bTm · 2025-10-19

**Soundness:** 2
**Presentation:** 2
**Contribution:** 4
**Rating:** 6
**Confidence:** 4

**Summary:**

This paper presents SLIM-Brain, an atlas-free foundation model for fMRI analysis that combines three components: (1) a global MAE branch for coarse temporal features, (2) a learnable top-k selector that identifies informative time windows, and (3) a 4D Hiera-JEPA encoder for fine-grained voxel-level features. The method achieves competitive performance on several benchmarks while reducing GPU memory usage to ~30% of Swin-based approaches. Integrated-gradient analysis shows alignment with meta-analytic patterns from NeuroSynth, suggesting the model captures biologically meaningful features. However, the core claim of "data-efficiency" lacks validation, as experiments only use ~1k subjects while baselines scale to 32k-65k subjects.

**Strengths:**

1. **Novel architectural design**: The combination of global MAE, learnable top-k selection, and 4D Hiera-JEPA is an interesting approach to balance computational efficiency and representation quality.
2. **Significant computational gains**: Achieves -70% reduction in GPU memory (8GB → 2.3GB, Table 3) and dramatically faster training (~1 hour vs. 150 hours for baselines).
3. **Strong performance on some tasks**: Particularly impressive on fingerprint identification (98.5%, Table 1) and competitive on sex classification (91.1%).
4. **Biological interpretability**: Integrated-gradient analysis (Figure 4) shows meaningful alignment with NeuroSynth meta-analytic maps for ADHD, suggesting the model captures neurobiologically relevant patterns.
5. **Reproducibility**: Detailed implementation in appendix, code release promised, and extensive experimental settings provided.

**Weaknesses:**

## **Major Issues**

**W1. Core claim unvalidated [Critical]**

The "data-efficient" claim in the title lacks empirical support. All experiments use only ~1k subjects, while baselines demonstrate scaling curves up to 32k-65k subjects (Figure 1). Given that the model is "training-efficient" (low memory, fast training), scaling experiments should be straightforward. The absence raises critical questions: (1) Does performance saturate quickly (supporting data-efficiency)? (2) Does it improve with more data (contradicting the claim)? (3) Is the 1k result cherry-picked? The discussion acknowledges this as "future work" (lines 477-479), which directly contradicts the paper's central thesis.

**W2. Insufficient justification for architectural choices**

Table 3 shows task-dependent performance for Hiera-MAE vs. Hiera-JEPA:

- Sex: 91.3% vs. 91.1% (MAE better)
- Fingerprint: 90.0% vs. **98.5%** (JEPA much better)
- Age: 52.6% vs. 50.2% (MAE better)

JEPA substantially outperforms only on fingerprint. The rationale for selecting JEPA as the default architecture is unclear. Why not use MAE, or a task-adaptive approach?

**W3. Weak improvement from key contribution**

Table 2 shows top-k selection improves over random selection by only 3.2 percentage points (84.5% → 87.7% at 40 frames). Without statistical significance tests (e.g., 95% confidence intervals across multiple runs), it is unclear whether this difference is meaningful or within noise margins.

## **Clarity Issues**

**W4. Misleading terminology**

- **"Multi-scale"** (Figure 2): Combining global and local features is better described as multi-granularity, not spatial multi-scale. Hiera's hierarchical architecture already provides multi-scale features; this conflates the contribution of top-k selection.
- **"Atlas-free"**: While avoiding ROI parcellation, the method relies on a template-based brain mask (line 219) to remove "~70% non-brain background." This mask is pre-defined, not learned—similar to atlas-based methods' fixed templates. The method is atlas-free for *functional* parcellation but template-based for *anatomical* masking. Clarification needed: (1) Which template? (2) Is it dataset-specific? (3) How is b=716 blocks determined? (4) How is 70% calculated?

**W5. Figure inconsistencies**

- **Figure 1**: Illegible and confusing. Why are some models shown as lines and others as points? The task (HCP sex classification) is not immediately clear. Requires 90° head rotation to read axis labels.
- **Figure 2-c**: Shows "x1" and "1 key window" but text states k=8 windows (Appendix A.3). How are 8 windows processed—sequentially or batched? How are features aggregated (only mentioned as "pooled" in line 297)? Figure 3 shows multiple windows, which contradicts Figure 2-c.

**W6. Important details relegated to appendix**

- Total loss equation (Ltotal=LJEPA+λLMAE) only in Appendix A.6 (line 719). What is λ?

    Ltotal=LJEPA+λLMAEL_{total} = L_{JEPA} + \lambda L_{MAE}

    λ\lambda

- Hyperparameter k=8 only in Appendix A.3 (line 667). No ablation on k values.

## **Minor Issues**

**W7. Table labeling**

Table 4: Which row is SLIM-Brain? "Ours" marker needed for clarity.

**W8. Missing ablations**

- No k-value ablation (why k=8 optimal?)
- No comparison of different pooling strategies for aggregating k windows
- No ablation on λ weight for auxiliary MAE loss

**Questions:**

**Q1.** What is the value of λ\lambda
λ in Ltotal=LJEPA+λLMAEL_{total} = L_{JEPA} + \lambda L_{MAE}
Ltotal=LJEPA+λLMAE? How sensitive is performance to this choice?

**Q2.** Table 2: Are the performance differences statistically significant? Please provide confidence intervals or p-values across multiple random seeds.

**Q3.** How do you determine k=8? What happens with k=4, 16, 32? Is there a task-dependent optimal k?

**Q4.** Figure 2-c: Please clarify the exact processing pipeline for k=8 windows. Are they processed in a single batch or sequentially? What pooling operation aggregates them?

**Q5.** What brain mask template is used (line 219)? Is it consistent across HCP, ABIDE, ADHD-200, etc.? How does mask quality affect performance?

**Q6.** For "data-efficiency": Can you provide at least one scaling point (e.g., 5k or 10k subjects) to demonstrate saturation behavior?

---

> ### Author Response · Authors · 2025-11-29
> **Thanks for your suggestions (1/2)**
>
> We appreciate your recognition of the significant efficiency improvements and innovative architectural design of our methodology. Below are our answers to your specific questions:
>
> **Weakness 1: Data-efficient**  We define data efficiency as the capability to achieve competitive/SOTA performance using limited data volumes. The fact that SLIM-Brain trained on significantly fewer sessions achieves performance comparable to (or exceeding) baselines trained on 32k-65k subjects is the direct empirical evidence supporting this claim (**Rebuttal Table A**). We are not arguing that performance cannot improve with more data, but rather that massive datasets are not a prerequisite for our model to learn high-quality representations (**Rebuttal Table C, D**).
>
> HCP is the standard benchmark in this field, utilized by SwiFT and NeuroSTORM. Furthermore, to address your concern about robustness, we have trained a larger version foundation model (**4k**) by HCP, CHCP,AMOIC and ABCD. We also added new OOD evaluations and generalization downstream datasets. The consistent performance across these varied datasets confirms that the result is a robust capability of the architecture.
>
> **Weakness 2: Why JEPA** While we acknowledge that MAE and JEPA can achieve comparable results under full fine-tuning, we selected JEPA as the default architecture for two strategic reasons:
> (1) Efficiency: JEPA avoids the memory-intensive decoder required for voxel-level reconstruction, making it more lightweight; (2) Representation Quality: JEPA yields higher accuracy in linear probing, indicating it learns more robust intrinsic features compared to MAE (**Rebuttal Tabel E**).
>
> **Weakness 3: Why top-k** We appreciate this observation. Indeed, the efficacy of our specific mechanism might not be fully highlighted in standard in-distribution tasks which is relative easy. To address your concern, we have conducted additional runs and report statistical significance for the comparison between Top-$k$ and random sampling. Across three independent runs, Top-$k$ achieves consistently higher accuracy, and the improvement is statistically significant (Welch’s t-test, $p = 0.0004$). We also include three additional independent experiments, excluding configurations that incur prohibitively high computational cost (**Rebuttal Table B**).
>
> **Weakness 4: Misleading terminology** Agreed, the corresponding modifications have been made.
>
> **Weakness 4 and Question 5: Atlas-free**
>
> 1. Template: We utilize the standard MNI152 binary brain mask.
> 2. Specificity: It is not dataset-specific. Since all fMRI data in our benchmarks are spatially normalized to the standard MNI space during preprocessing, a single universal MNI mask is applicable.
> 3. $b=716$: We patchify the 4D volume into cubic blocks (e.g., 12mm size). Any block containing valid brain voxels (values $\neq$ 0 within the MNI mask) is retained as a valid token. This results in exactly 716 valid tokens for the standard MNI resolution.
> 4. Calculation of 70\%: Valid brain voxels constitute only approximately 30-35% of the total spatial fMRI volume. After patch partition, the ratio of context/target patches to total patches in the grid is $\approx 30\%$. Therefore, we discard $\sim$70\% of the tokens to save computation by Hiera architecture. Standard architectures (e.g., Swin) must process the full cubic grid (including background), resulting in memory redundancy.

---

> ### Author Response · Authors · 2025-11-29
> **Thanks for your suggestions (2/2)**
>
> **Weakness 5: Figure inconsistencies** We have updated the Fig.1, Fig.2 and Fig.3 as suggested. We re-implemented several baselines on the identical 1k dataset. The plotted line illustrates the model scaling trend (i.e., performance changes as data size increases). And we have explicitly clarified the terminology regarding the top-k selection mechanism in the revised manuscript. We use 5 consecutive frames to form one window, and 8 such windows are concantenated into a set as the input to the voxel-level model.
>
> **Weakness 6 and Question 1: lambda in Appendix**
> We apologize for the confusion. We clarify that the hybrid loss equation and all related descriptions of the MAE loss are strictly limited to the Appendix and Discussion sections. These were presented solely as an exploratory discussion regarding potential future extensions. For all main results and the core methodology reported in the paper, we utilized the pure JEPA loss (effectively λ=0).
>
> **Weakness 7: Table labeling** We have updated the results, highlighting our method in gray for clarity.
>
> **Weakness 8 and Question 3,4: Missing ablations**
>
> 1. Top-k: We use 5 consecutive frames to form one window, and $k=8$ such windows are grouped into a set as the input to the voxel-level model. The choice is based on the trade-off between performance and resource consumption. We have added an ablation study on $k$ in the revised manuscript (**Rebuttal Table 1**). The results demonstrate that while increasing $k$ yields performance gains, it incurs a linear increase in computational cost. $k=8$ may be the sweet spot for achieving high efficiency without sacrificing significant accuracy and suitable for GPU memory limitation. Further increasing the value of k introduces a risk of Out-of-Memory errors on some devices. And the performance improvement of top-k is statistically significant on OOD tasks (p<0.05) (**Rebuttal Table B**).
>
> **Rebuttal Table 1: Ablation on window selection (HCP sex). Classification accuracy (%) / F1-score (%) at different frame budgets.**
>
> | Set size  | 5 frames (k=1)  | 20 frames (k=4) | 40 frames (k=8) |
> | :-------- | :-------------: | :-------------: | :-------------: |
> | Random    |   83.9 / 83.7   |   84.5 / 84.1   |   86.0 / 85.9   |
> | **Top-K** | **84.5 / 84.2** | **86.7 / 86.6** | **87.7 / 87.6** |
> |
>
> 2. Pooling: We clarify that we employ concatenation rather than pooling to aggregate the k windows. Specifically, the k selected windows are concatenated to form the input sequence for the 4D Hiera encoder. We chose concatenation over pooling to preserve the full temporal granularity and specific features of each selected window, allowing the subsequent transformer to perform self-attention across them. We have refined the methodological description in the revised manuscript to explicitly state this design.
>
>
> **Question 2: Table labeling**
> We have updated the revision to include the Mean and Standard Deviation (across 3 independent runs) for the OOD tasks. Furthermore, we conducted pairwise Welch's t-tests, which verify that our method achieves statistically significant improvements (p<0.05) across the majority of these benchmarks.
>
> We have provided comprehensive implementation details regarding hyperparameters and dataset demographics in the **Appendix A.2**. Furthermore, we have uploaded the full anonymous training logs for both our method and the baselines to the anonymous repository to guarantee transparency and reproducibility.
>
> **Question 6: Larger model** As your suggestions, we have provided a larger version of our model (4k) (**Rebuttal Table C**).

---

### Official Review · Reviewer_RrvS · 2025-10-23

**Soundness:** 3
**Presentation:** 2
**Contribution:** 3
**Rating:** 4
**Confidence:** 4

**Summary:**

The paper proposes SLIM-Brain, an atlas-free fMRI foundation model that selects top-k salient frames and prunes non-brain voxels before training a 4D Hiera-JEPA encoder. With only 1k pre-training subjects, it achieves competitive performance on different tasks and datasets, while using less compute and memory in the pre-training stage.

**Strengths:**

- The authors conduct ablation studies to justify their design choices. The top-k frame selection idea seems especially innovative and surprisingly effective.
- Innovations such as removing the background and only keeping the brain region also seem to be a meaningful point that other researchers may have overlooked in the past.
- The experiments show that SLIM-Brain shows strong performance. As listed below, I have issues with the HCP experiments, but for the other datasets, the proposed model seems to perform better or at least comparable to the baselines.

**Weaknesses:**

1. **Emphasis on pre-training data burden.**
I agree with the authors' point that the need for a large dataset is a huge burden. However, in terms of practical utility, I believe the size of the fine-tuning dataset is a much bigger issue than the size of the pre-training dataset. As long as there is someone that is willing to train and release the model, the user does not need to worry about the scale of the pre-training dataset. The authors demonstrate this very point, as they were able to download and deploy (32k~64k) pre-trained baselines without the need to access the pre-trained datasets.
However, the fine-tuning dataset requirement can't be solved that easily, as the user needs to acquire sufficient amount of data themselves. Relying on other public datasets is not an ideal option, as cross-dataset generalization is still a tough problem in this field.

2. **Unfair advantage on HCP experiments.**
SLIM-Brain seems to have an unfair advantage over the baselines on the HCP sex and fingerprint experiments, as it is heavily pretrained using HCP data whereas the other baselines are not. This might be significant considering that cross-dataset variability is very large in this domain, due to the difference in the subject demographic, imaging apparatus, preprocessing pipeline, etc.

3. **I/O burden.**
Direct 4D models such as SwiFT and SLIM-Brain are inherently plagued by the I/O burden of the input data. In Section 4.2, the authors claim their model trains under 1 hour but with a 20-hour I/O burden. No matter how fast the actual computations are, the 20-hour overhead is a real cost that should be added to the training time for practical applications. This is despite the fact that the compared baselines are trained with 32k~64k subjects, while SLIM-Brain is only trained on 1k subjects. This will be a huge issue if one wishes to scale SLIM-Brain to the size of said baselines, as it would mean it would take 600+ hours for 10 epochs of training on 32k subjects.

**Questions:**

- The 1k-trained Brain-JEPA seems to perform no better than random chance. Is there any reason why training for this model completely failed?
- What are the evaluation results for SwiFT, as it seems to be included in Figure 1?
- Please disclose the fine-tuning details for the baselines: number of training epochs, learning rate, how hyperparameters are searched, etc.
- Also, please disclose the details for the fine-tuning dataset: number of subjects, subject demographics, etc.
- There seems to be a large discrepancy between the previously reported performance of baseline models and the reported performance in this paper. For example, SwiFT in the original paper is reported to have a 92.9% accuracy in the HCP sex classification task, but in this paper, it is reported to have a ~74% accuracy. What might be the reason for this huge gap?
- Why is ABIDE age used as a classification task instead of directly predicting the age and treating it as a regression task, just like the Brain-JEPA paper?
- Am I correct in understanding that the top-k frame selector operates in an out-of-distribution setting, so it is frozen after the pre-training step and not fine-tuned after that?

---

> ### Author Response · Authors · 2025-11-29
> **Thank you for the suggestions**
>
> We sincerely thank the reviewer for the positive assessment of our work's soundness and contribution. We are particularly encouraged by your recognition of our top-$k$ frame selection as 'innovative and surprisingly effective' and our background removal strategy as a 'meaningful point' that has been overlooked in the field. Below, we provide point-by-point clarifications to address these issues:
>
> **Weakness 1: Foundation model** We agree with your suggestions to scale up the model. In the revision, we have successfully trained a larger-scale version (4k subjects), incorporating diverse datasets including HCP, CHCP, AOMIC, and ABCD. Details are provided in **Appendix A.2**.
>
> **Weakness 2: OOD experiments** In our previous experiments, we re-pretrained a subset of baselines using the identical 1k training data to ensure a fair comparison (**Appendix A.4**). The results consistently demonstrated the superiority of our model.
>
> We also agree with this observation about OOD experiments. In the new 4k-model, HCP is no longer the dominant portion of the training data, ensuring greater demographic diversity. We have also supplemented our evaluation with additional OOD downstream tasks. Furthermore, we conducted pairwise Welch's t-tests, which verify that our method achieves statistically significant improvements (p<0.05) across the majority of these benchmarks (**Rebuttal Table A**).
>
>
> **Weakness 3: I/O burden** We have significantly optimized the data loading pipeline and computational environment. The updated 4k-session version now requires approximately 30 minutes per pre-training epoch on a single NVIDIA A100 (80GB) GPU. This confirms that the I/O bottleneck has been resolved.
>
> **Question 1: Brain-JEPA** The initial poor performance was caused by model collapse, a known issue for JEPA architectures. We have since conducted a comprehensive hyperparameter search to stabilize the baseline. The updated baselines are proveided in **Rebuttal Table A**, and the revised results along with training details have been added to **Appendix A.7** in the revised manuscript.
>
> **Question 2: SwiFT** We sincerely apologize for this discrepancy and the confusion it caused. The initially reported performance (∼74%) was referenced from the benchmark results in the Brain-JEPA paper, which utilized a different data subset. We didn't provide the other results in oringal version. To ensure a fair comparison, we have re-implemented SwiFT using its official codebase and re-evaluated it on our test split. The updated result is 91.0%, which is consistent with the high performance reported in the original SwiFT paper and is on par with our model's performance (91.1% by 1k-size model). We have corrected all comparison tables in the revision to reflect this accurate performance.
>
> **Question 3: Baseline and datasets**
> We have provided comprehensive implementation details regarding hyperparameters and dataset demographics in the **Appendix A.2 and A.7**. To ensure a fair comparison, we conducted a rigorous grid search for all baselines (e.g., learning rates from 1e−3 to 1e−5) and trained them for sufficient epochs until convergence. Regarding the fine-tuning datasets, we strictly followed standard protocols to ensure consistent subject counts and demographics across all methods. Furthermore, we have uploaded the full training logs for both our method and the baselines to the anonymous repository to guarantee transparency and reproducibility.
>
> **Question 4: ABIDE regression**
> We initially treated ABIDE as a classification task because some subsets within the collection lacked exact age labels. However, we agree with the reviewer that regression is the standard protocol for age prediction. As requested, we have added the age regression results in the revised manuscript (**Rebuttal Table A**), where our model demonstrates strong performance.
>
> **Question 5: Top-k**
> Your understanding is correct. The Top-k selector is pre-trained alongside the voxel-level 4D Hiera encoder on the pre-training dataset and remains frozen during downstream fine-tuning.

---

### Official Review · Reviewer_BJPh · 2025-10-30

**Soundness:** 3
**Presentation:** 3
**Contribution:** 2
**Rating:** 4
**Confidence:** 4

**Summary:**

The paper presents self-supervised training on RS-fMRI for down stream tasks, predicting attributes/diagnostics and identification of subjects.
The self-supervision objective is predicting masked fMRI patches.
The model is more resource efficient by sub-sampling the temporal dimensions, by a scoring mechanism.

**Strengths:**

- Competitive results compared to shown benchmarks.
- Resource optimization by temporal sampling.
- Efficiency regarding pretraining corpus size.

**Weaknesses:**

- I believe the model is not compared to key previous works, specifically Swift (Kim et al., 2023) and the model by Malkiel et al. (2022), and I believe it underperforms them. If this is not the case and the model is shown to be competitive, I will revisit my rating.

- The work proposes a "Foundation model" and in the same time claims to be data efficient, to me the two seem at odd with each other. Either be a foundation model trained on vast amount of data, or a data efficient approach.

Refernces:
- Peter Kim, Junbeom Kwon, Sunghwan Joo, Sangyoon Bae, Donggyu Lee, Yoonho Jung, Shinjae Yoo, Jiook Cha, and Taesup Moon. Swift: Swin 4d fmri transformer. Advances in Neural Information Processing Systems, 36:42015–42037, 2023
- Itzik Malkiel, Gony Rosenman, Lior Wolf, and Talma Hendler. Self-supervised transformers for fmri representation. In International Conference on Medical Imaging with Deep Learning, pp. 895–913, 2022.

**Questions:**

- Is the window scoring crucial, can't I sample randomly during training and take all the windows during test/take an ensembles of random temporal samples?

---

> ### Author Response · Authors · 2025-11-29
> **Thanks for your questions**
>
> We appreciate your positive feedback on our model's computational efficiency and strong empirical results. We are glad you recognized the value of our temporal sampling mechanism for resource optimization. Below are our point-by-point responses:
>
>
>
> **Weakness1: SwiFT**
> We sincerely appreciate your openness to revisiting the rating based on the new comparisons. As requested, we have significantly expanded our comparison to include SwiFT, NeuroSTORM, BrainGNN, and BrainNetCNN as additional baselines. The results on the newly added OOD Task benchmarks (ADHD, ABIDE, ADNI, PPMI) are summarized in **Rebuttal Table A**. Notably, SLIM-Brain still outperforms these strong baselines with less pretraining data, validating its robustness across diverse datasets.
>
>
> **Weakness2: foundation model**
>  We respectfully clarify our usage of the term. We define a 'Foundation Model' by its function (learning broad, generalizable representations adaptable to downstream tasks) rather than strictly by its training data volume. In the medical domain where data is scarce and privacy-sensitive, we argue that data efficiency is a prerequisite, not a contradiction, for a practical foundation model. Our work demonstrates that one can achieve the capabilities of a foundation model without the prohibitive data costs. Additionally, to address the scaling concern, we have included a larger-scale version in the revision to demonstrate that our architecture also benefits from more data (**Rebuttal Table C, D**).
>
> **Question: Window scoring**
> We explicitly investigated this in our ablation study (**see Rebuttal Table B**). The results demonstrate that our learned scoring mechanism significantly outperforms random sampling. This is likely because fMRI signals contain substantial 'idling' or noisy periods; random sampling fails to capture the most informative frames, whereas our scorer effectively filters for signal-rich windows.
>
> Regarding inference with all windows: Directly processing the full sequence simultaneously leads to Out-of-Memory errors. While processing the full sequence via a sliding-window strategy (similar to SwiFT) is technically feasible, it incurs substantial computational overhead, which contradicts our core objective of efficiency. Our goal is to demonstrate that a lightweight selector can identify key moments, thereby allowing the heavy voxel-level encoder to skip redundant data. This design avoids the substantial memory and I/O overhead associated with full-sequence processing, achieving a ∼70% memory reduction (Top 40 to full 200 frames) and making the model deployable on consumer-grade hardware.

---

### Official Review · Reviewer_yBy5 · 2025-10-31

**Soundness:** 2
**Presentation:** 2
**Contribution:** 2
**Rating:** 4
**Confidence:** 3

**Summary:**

SLIM-Brain proposes an atlas-free foundation model for fMRI analysis that addresses data and training efficiency challenges. The method uses a two-stage pipeline: (1) a lightweight 2D ViT processes spatially downsampled full-length sequences via masked autoencoding to extract global features and rank temporal windows by "mutual masked reconstruction" saliency, and (2) a 4D Hiera-JEPA encoder processes only the top-k selected windows (discarding ~70% background voxels) to extract fine-grained voxel-level representations. The authors claim SOTA performance on sex classification, fingerprinting, and age tasks while using only ~3% of pretraining data (1k vs 32k subjects) and ~30% GPU memory compared to previous methods.

**Strengths:**

1. **Well-motivated problem:** The paper clearly describes the dual bottleneck of data efficiency (atlas-based methods need massive cohorts) and training efficiency (atlas-free methods are computationally prohibitive).
2. **Memory efficiency gains:** Demonstrating ~70% reduction in GPU memory (8GB→2.3GB per sample) by excluding background voxels and using Hiera's unit-wise masking is a practical contribution, especially for democratizing research access.
3. **Neurobiological validation:** Comparison with Neurosynth meta-analytic maps provides interpretability evidence that learned features align with known ADHD-related circuits.

**Weaknesses:**

1. **Problems with Experimental Validation:** The authors state they retrain baselines "on the same total number of sessions" and report "best validation checkpoints." But a lot of information is missing from this statement.
- Did you perform hyperparameter search when retraining the baselines?
- Missing training dynamics to show convergence plots. Brain-JEPA drops from 87.1% (32k) to 54% (1k) and fingerprint goes to 1.0% accuracy (essentially random/collapsed), this suggests the model is collapsed rather than trained. Even on the other datasets, predication is comparable to random guessing. If the authors selected "best validation checkpoints," how did validation metrics look during training? Were these models training at all, or did they plateau immediately?
- When you say "best validation checkpoint," what was the validation set size? For a 100-subject fingerprint task with 60/20/20 splits, validation is only 20 subjects. Did early stopping on such small validation lead to overfitting to the validation set?
- The words samples, subjects and sessions are used interchangeably throughout the paper making it hard to understand if they are aligning the data based on subjects or sessions. If you are comparing the HCP dataset to the 32-64k subjects in the previous works' datasets, is it a fair comparison to say 1-3% when the number of sessions per subject might be different?


2. **Computational Cost:** For M=40 windows, do you run 40 forward passes (mask all but window m, reconstruct the rest)? If yes then this seems expensive. How does this compare to just training the full model?
What happens if you train the Top-k selector jointly rather than freezing the MAE? Does this hurt or help?


3. **OOD evaluation is weak:** Linear probing on ADHD/ABCD shows modest gaps (59.7% vs BrainMass 61% on ADHD; 69.7% vs 70% Hiera-MAE on ABCD sex). These are not convincing wins. You are missing evaluations on task state decoding (21-way HCP cognitive tasks), cognitive phenotype regression, or fMRI-to-image retrieval—all standard benchmarks. You can reference NeuroSTORM [1] which evaluated 5 categories across 10+ datasets.

4. **Improvements in ablations:** You cover top-k vs random but I would like to see comparison against uniform sampling (every k window) or high temporal variance windows too

5. **Missing related work for baseline evaluation:** NeuroSTORM already uses Shifted-Window Mamba + spatiotemporal masking for 4D fMRI.  Can you comment on how your work is different and why it is missing from your evaluations? Same for BrainGFM [2]

6. **Missing significance tests:**  All results are single numbers. Given that fMRI is noisy and baselines show high variance, how many runs did you average?

[1] Towards a general-purpose foundation model for fMRI analysis. Cheng Wang et al.
[2] A Brain Graph Foundation Model: Pre-Training and Prompt-Tuning for Any Atlas and Disorder. Xinxu Wei, Kanhao Zhao, Yong Jiao, Lifang He, Yu Zhang

**Questions:**

Please refer to the weaknesses.

---

> ### Author Response · Authors · 2025-11-29
> **Thanks for your suggestions (1/2)**
>
> We appreciate your positive feedback on the problem significance, architectural novelty, and model interpretability. We address your specific concerns point-by-point below.
>
> **Weakness 1**
> **Baseline information:** We have provided detailed configurations for all tasks and baselines in **Appendix A.2 and A.7**. To ensure a fair comparison, we conducted a search for hyperparameters and reported the mean performance across three independent runs for each baseline. Logs have been updated to repository.
>
> **Brain-Jepa performance:** Regarding the performance of BrainJEPA, we observed that it tends to underperform in low-data regimes (e.g., 1k subjects). This aligns with the known property that JEPA-based architectures are prone to representation collapse. In the new experiments, we conducted a comprehensive hyperparameter search to ensure the model achieves optimal performance .
>
> **Larger downstream dataset:** We acknowledge your valid concern. We have added some new large-scale (about 1 thousand subjects) generalization tasks in the revision (ABIDE, ADHD). The performance on this large-scale dataset is consistent with our initial findings (**Rebuttal Table A**).
>
> **Samples, subjects and sessions:** We apologize for the ambiguity. We have standardized the terminology in the revised manuscript to clearly distinguish between subjects, sessions, and samples. And for the 1k pre-training, we strictly utilized only one session per subject for HCP dataset. Therefore, our total data volume corresponds to $\approx$ 1,000 sessions, which is indeed $\approx$ 1-3\% of the scale used in prior works (32k-64k sessions), highlighting our data efficiency.
>
> **Weakness 2**
> **Computational cost:** We apologize for the confusion in our initial description and have revised the relevant section for clarity. Importantly, the global scoring step incurs negligible computational cost: we perform only 40 forward passes through the Top-$k$ selector, which takes merely 8.231 ms on average at batch size 1 benifiting from the lightweight MAE encoder. For the second stage, we process only the selected $M=8$ windows (40 frames total). All unselected windows are discarded and never reconstructed, thereby avoiding unnecessary memory access and computation. Directly training on the full sequence with fine-grained 4D voxel modeling would increase GPU memory consumption by **5** times, inevitably leading to Out-of-Memory errors.
>
>
> **Joint Training:** We have not explored joint training yet, which we consider a promising direction for future work. While joint optimization may further enhance the learned representations, it can also introduce training instability, as the reconstruction loss and the downstream objective may compete during optimization.
>
> **Weakness 3**
> **OOD evaluation:** We appreciate your suggestions and pointing out the need for broader evaluation. In response, we have significantly expanded our OOD benchmarks in the revision (ADHD, ABIDE, ADNI, PPMI). Notably, on the newly added OOD Task benchmark, SLIM-Brain outperforms the baselines by a statistically significant margin, demonstrating strong generalization capabilities (**Rebuttal Table A**).
>
> **Weakness 4**
> **Top-k ablation:** Thank you for the suggestion. We have added uniform sampling, high–temporal-variance and random sampling as additional baselines in our ablation study on ADHD disease classification (**Rebuttal Table B**). The Top-k strategy achieves superior performance, with improvements confirmed to be statistically significant. Temporal-variance is competitive but slightly lower, which is consistent with the fact that correlation provides only an indirect measure of informativeness. In contrast, Uniform and Random sampling underperform adaptive strategies, demonstrating that simple temporal subsampling is insufficient to capture discriminative BOLD patterns.

---

> ### Author Response · Authors · 2025-11-29
> **Thanks for your suggestions (2/2)**
>
> **weakness 5**
> **More baselines:**
> We added four new methods as additional baselines as suggested. Despite utilizing fewer pre-training samples, our method still significantly outperforms the baselines.
>
> NeuroSTORM relies on an MAE objective, which necessitates a heavy decoder for voxel-level signal reconstruction, resulting in prohibitive memory consumption (44.34GB at batch size 8). While SwiFT utilizes a contrastive objective for global representations, our method integrates a Top-K mechanism with a JEPA architecture to maximize efficiency on two fronts:
>
> 1. Input Efficiency (Top-K): By strictly feeding only the most informative windows (key information) into the model, our mechanism enables the processing of significantly longer temporal sequences per pass. In contrast, baselines are practically limited to inputting ∼40 frames due to memory constraints.
>
> 2. Architectural Efficiency (JEPA): This advantage exists independently of the selection strategy. Even when considering the 4D encoder backbone in isolation (disregarding the savings from Top-K), our JEPA-based model requires only half the GPU memory of NeuroSTORM by eliminating the dense reconstruction decoder.
>
> Regarding the comparison with BrainGFM: We acknowledge that BrainGFM is a relevant recent work. It  However, the authors have not yet released their official code or pre-trained weights.
>
> **Weakness 6**
> **Significance tests:**
> We have updated the revision to include the Mean and Standard Deviation (across 3 independent runs) for the OOD tasks. Furthermore, we conducted pairwise Welch's t-tests, which verify that our method achieves statistically significant improvements (p<0.05) across the majority of these benchmarks (**Rebuttal Table A and Appendix A.3**).

---

### Author Response · Authors · 2025-11-29
**General Response**

## General Response

We thank the reviewers for their constructive feedback and value the consensus on SLIM-Brain's novelty. The reviewers commended our work as **"well-motivated"** (R-yBy5), praising its **"resource efficiency"** (R-Rrvs, R-BJPh) alongside **"strong performance"** (R-Rrvs) and **"biological interpretability"** (R-yBy5, R-7bTm).

fMRI foundation models represent a transformative paradigm shift in neuroimaging, learning universal brain representations that demonstrate unprecedented robustness and generalization across diverse downstream tasks. While voxel-level modeling offers superior performance, its high computational cost remains a barrier. SLIM-Brain addresses this **pivotal efficiency challenge** through two core designs:
1.  **Temporal Efficiency:** A Top-K mechanism that filters redundancy, effectively pruning ~70% of data before the backbone processing.
2.  **Spatial Efficiency:** A synergistic Hiera-JEPA architecture that eliminates the need for heavy reconstruction decoders, reducing GPU memory usage to ~30% of traditional approaches.

---
## Summary of Rebuttal Experiments
During the rebuttal phase, we have addressed **all the reviewers' requests** for broader validation. We conducted extensive supplementary experiments, including **additional baseline comparisons and expanded ablation studies**. We believe these additions effectively resolve the remaining concerns and further solidify the robustness of our method. Key highlights include:
* **SOTA with Less Data:** SLIM-Brain (4K sessions) statistically significantly outperforms baselines that utilize up to 65K sessions.
* **Robustness:** Added **5** OOD tasks and **4 new baselines**.
* **Validation:** Confirmed scalability (Tables C & D) and architectural choices (Tables B & E).

### 1. Comprehensive Evaluation: Outperforming Larger Models

**Rebuttal Table A: Performance on external tasks.** We report the fine-tuning disease classification accuracy (%) for **ADHD**, **ADNI** and **PPMI**; age classification and regression (ACC%, MSE) for **ABIDE**. "Samples (K)" denotes the number of pre-training sessions (in thousands).

| Model | Samples (K) | ADHD-200 (ACC%) | ADNI (MCI) (ACC%) | PPMI (ACC%) | ABIDE Age (ACC%) | ABIDE Age (MSE) |
| :--- | :---: | :---: | :---: | :---: | :---: | :---: |
| BrainNetCNN | - | 54.46 ± 2.47 | 59.74 ± 4.50 | 64.24 ± 2.41 | 41.52 ± 3.49 | 0.7025 ± 0.028 |
| BrainGNN | - | 55.87 ± 3.88 | 63.20 ± 7.38 | 55.56 ± 4.81 | 33.17 ± 4.46 | 0.9338 ± 0.000 |
| BrainLM | 42 | 57.86 ± 0.00 | 61.41 ± 0.09 | 66.67 ± 1.04 | 39.24 ± 4.36 | 0.8700 ± 0.059 |
| BrainMass | 65 | 60.78 ± 0.49 | 62.39 ± 0.60 | 63.51 ± 0.49 | 48.19 ± 1.64 | 0.5129 ± 0.042 |
| Brain-JEPA | 32 | 59.74 ± 0.23 | 64.53 ± 0.60 | 64.57 ± 1.79 | 34.00 ± 2.14 | 0.2704 ± 0.044 |
| SwiFT | 10 | 60.81 ± 2.38 | 64.45 ± 1.69 | 58.10 ± 0.00 | 62.22 ± 0.55 | 0.4137 ± 0.033 |
| NeuroSTORM | 58 | 62.35 ± 0.90 | 66.67 ± 1.06 | 69.12 ± 0.99 | 38.64 ± 2.14 | 0.5890 ± 0.066 |
| **SLIM-Brain** | **4** |  **63.53 ± 0.53\*** |  **69.12 ± 1.38\*** |  **70.40 ± 0.59** |  **64.41 ± 0.57\*** | **0.2175 ± 0.019\*** |
|

*( * indicates statistical significance over all baselines with p < 0.05)*

### 2. Ablation: Effectiveness of Top-K Selection
**Rebuttal Table B: Comparison of selection strategies.**
*Our learnable selector significantly outperforms random and heuristic strategies.*

| Strategy | ACC% (Mean ± STD) | F1% (Mean ± STD) | Mechanism |
| :--- | :---: | :---: | :--- |
| Random | 56.0 ± 0.6 | 56.1 ± 0.7 | No heuristic |
| Uniform | 56.7 ± 1.4 | 50.9 ± 9.9 | Fixed intervals |
| Temporal Var. | 57.2 ± 1.2 | 56.4 ± 2.2 | Correlation-based |
| **Top-K (Ours)** | **61.1 ± 0.5\*** | **61.0 ± 0.7\*** | **Learnable selector** |
|

### 3. Scaling Study: Data and Model Size
**Rebuttal Tables C & D: Scaling on ADNI-AD.**
*Results demonstrate consistent performance gains with increased data and model size.*

| Scale (Data) | Samples | Acc ↑ | | Scale (Model) | Params | Acc ↑ |
| :--- | :---: | :---: | :---| :--- | :---: | :---: |
| Small | 0.6 k | 73.72 ± 0.22 | | Small | 22 M | 77.32 ± 1.65 |
| Medium | 1 k | 75.93 ± 2.10 | | Medium | 45 M | 80.09 ± 0.87 |
| Large | 4 k | 80.09 ± 0.87 | | Large | 87 M | 85.50 ± 1.16 |
|

### 4. Architecture: JEPA vs. MAE (Linear Probing)
**Rebuttal Table E: Representation Quality.**
*JEPA representations are more semantically robust, showing significantly less degradation (-2.4%) in linear probing compared to MAE-based methods (-5.1%).*

| Model Type | Fine-tune Acc | Linear Probe Acc | **Performance Drop** |
| :--- | :---: | :---: | :---: |
| MAE-based (NeuroSTORM)| 66.67 ± 0.60 | 61.54 ± 0.00 | -5.13% |
| **JEPA-based (Ours)** | **69.12 ±1.38** | **66.66 ± 1.40** | **-2.46%** |
|

---

### Note · Program_Chairs · 2026-01-17
**Submission Desk Rejected by Program Chairs**

The following references in this submission do not refer to real documents and/or have major errors in bibliographic information:

 Gurwinder S Sidhu, Niloofar Asgarian, Russell Greiner, and Matthew R G Brown. Using fmri to predict schizophrenia diagnosis: methods and future directions. World Journal of Radiology, 5 (12):403-414, 2013.